# Integrative Metabolome and Transcriptome Analysis Reveals the Regulatory Network of Flavonoid Biosynthesis in Response to MeJA in *Camellia*
*vietnamensis* Huang

**DOI:** 10.3390/ijms23169370

**Published:** 2022-08-19

**Authors:** Heqin Yan, Wei Zheng, Yong Wang, Yougen Wu, Jing Yu, Pengguo Xia

**Affiliations:** 1Key Laboratory for Quality Regulation of Tropical Horticultural Plants of Hainan Province, College of Horticulture, Hainan University, Haikou 570228, China; 2Ministry of Education Key Laboratory for Ecology of Tropical Islands, College of Life Sciences, Hainan Normal University, Haikou 571158, China; 3Key Laboratory of Plant Secondary Metabolism and Regulation of Zhejiang Province, College of Life Sciences and Medicine, Zhejiang Sci-Tech University, Hangzhou 310018, China

**Keywords:** *Camellia vietnamensis* Huang, flavonoids, metabolomics, transcriptome

## Abstract

Flavonoids are secondary metabolites widely found in plants, which perform various biological activities, such as antiinflammation, antioxidation, antitumor, and so on. *Camellia vietnamensis* Huang, a species of oil-tea *Camellia* tree, is an important woody oil crop species widely planted on Hainan Island, which provides health benefits with its high antioxidant activity and abundant flavonoid content. However, very little is known about the overall molecular mechanism of flavonoid biosynthesis in *C**. vietnamensis* Huang. In this study, methyl jasmonate (MeJA) is used as an inducer to change the content of secondary metabolites in *C. vietnamensis*. Then, the potential mechanisms of flavonoid biosynthesis in *C. vietnamensis* leaves in response to MeJA were analyzed by metabolomics and transcriptomics (RNA sequencing). The results showed that metabolome analysis detected 104 flavonoids and 74 fatty acyls which showed different expression patterns (increased or decreased expression). It was discovered by KEGG analysis that three differentially accumulated metabolites (cinnamaldehyde, kaempferol and quercitrin) were annotated in the phenylpropanoid biosynthesis (ko00940), flavonoid biosynthesis (ko00941), and flavone and flavonol biosynthesis (ko00944) pathways. In the transcriptome analysis, 35 different genes involved in the synthesis of flavonoids were identified by MapMan analysis. The key genes (*PAL*, *4CL*, *CCR*, *CHI*, *CHS*, *C4H*, *FLS*) that might be involved in the formation of flavonoid were highly expressed after 2 h of MeJA treatment. This study provides new insights and data supporting the molecular mechanism underlying the metabolism and synthesis of flavonoids in *C. vietnamensis* under MeJA treatment.

## 1. Introduction

*Camellia vietnamensis* Huang, a species of oil-tea *Camellia* tree from Hainan Island, is more suitable for a tropical climate and has a higher content of active ingredients in the oil [1,2]. Its oil is rich in beneficial ingredients, such as the flavonoids, triterpenes saponin, squalene, vitamin E, and β-amyrin, and is used as a traditional Chinese medicine with multiple functions, including antioxidative, free radical scavenging, antibacterial and anticancer activities [3]. Flavonoids, as a natural polyphenol group in plants, widely exist in nature, including flavones, flavanones, isoflavones, and anthocyanidins [4,5]. Flavonoids have been reported to have various effects, such as antioxidant, anti-inflammatory, anticancer, antitumor, and protecting gastric mucosa functions [6]. Therefore, flavonoids are widely applied in the food, cosmetics, and medicine industries and have gained increasing attention [6,7].

The central pathway of flavonoid biosynthesis is conserved in plants [8]. Up to now, enzymes involved in flavonoid biosynthesis have been well characterized in *Vitis vinifera* [9], *Zea mays* L. [10], *Arabidopsis thaliana* L. [11], and *Fagopyrum tataricum* [12], and so on. Flavonoid biosynthesis involves both the phenylpropanoid metabolic pathway and the flavonoid biosynthetic pathway [13]. The flavonoid synthesis pathway begins with the catalysis of the precursor phenylalanine by phenylalanine ammonia lyase (*PAL*), followed by the production of the chalcone by trans cinnamate 4-monooxygenase (*C4H*) [6]. Chalcone synthase (*CHS*) is the first committed enzyme in the biosynthesis of all flavonoids [8], which catalyzes p-coumaroyl-CoA to generate naringenin chalcone or phloretin [6]. The high expression of the *CHS* gene enhances flavonoid accumulation in Perilla frutescens leaves by metabolomic and transcriptomic analysis [14]. Chalcone isomerase (*CHI*) catalyzes the isomerization of naringenin chalcone to flavanone naringenin [8]. Then flavanone naringenin acts as the substrate to produce flavones (such as flavanones, dihydroflavonols, and anthocyanins) by the action of various enzymes which include flavonoid 3-hydroxylase (*F3H*), flavonoid 3’-hydroxylase (*F3’H*), flavonoid 3’5’-hydroxylase (*F3’5’H*), flavonol synthase (*FLS*), anthocyanidin synthase (*ANS*), anthocyanin reductase (*ANR*) and so on [6,15]. In addition, several transcription factors (TFs) have also been shown to be involved in flavonoid biosynthesis. Overexpression of MdWRKY11 could promote the expression of *F3H*, *FLS*, *DFR*, *ANS*, and *UFGT*, resulting in increased accumulation of flavonoids and anthocyanin in apple calli [16]. However, the functions of genes related to flavonoid synthesis in *C. vietnamensis* have seldom been studied.

At present, multiomics combined analysis has greater advantages in comprehensively explaining the regulatory mechanisms of plant stress responses at different molecular levels by associating the changes in metabolites with gene expression [17,18]. Many scholars have conducted preliminary investigations into flavonoids synthesis pathways of a variety of plants by transcriptome and the metabolome, such as *Entada phaseoloides* [6], *Camellia sinensis* [19,20], *Eutrema salsugineum* [8], *Camellia oleifera* [21], and so on. As a well-known exogenous inducing factor, MeJA is widely employed to produce bioactive compounds and can affect secondary metabolite content is now a hot spot of research and has been largely validated on other species, such as *C. sinensis* [19] and *Carthamus tinctorius* L. [22]. MeJA could affect the up-regulation and down-regulation of genes in the flavonoid metabolic pathway in order to promote the production of flavonoid metabolites [22,23]. Consequently, MeJA treatment can be used to identify the genes involved in the biosynthesis of flavonoids. This method has been widely used in preliminary screening of genes related to the biosynthesis of secondary metabolites so far. Therefore, it is meaningful and feasible to analyze the biosynthetic pathways of flavonoids under MeJA treatment via both the transcriptome and the metabolome.

Secondary metabolites in *C. vietnamensis* exhibit various bioactivities, which can improve the quality, stability and health effects of *Camellia* seed oil. By combining proteomics and transcriptomics, our research team previously revealed the accumulation patterns of flavonoids and fatty acids in the seeds of oil-tea *Camellia* trees in Hainan at different developmental stages [24]. However, the biosynthetic processes of flavonoids in *C. vietnamensis* are not clear, and it is still difficult to explain the response mechanisms to biotic and abiotic stresses at the physiological and molecular levels. To this end, using ultra high-performance liquid chromatography with quadrupole time-of-flight mass spectrometry (UHPLC-QTOF/MS), we screened the metabolic expression profiles and flavonoid metabolites at different time points of MeJA treatment. Differentially accumulated transcripts of *C. vietnamensis* in response to MeJA signaling were determined by transcriptome analysis. Then, we constructed a gene coexpression network based on weighted gene coexpression network analysis (WGCNA) and combined it with correlation analysis to further screen the core key genes and metabolites. Based on previous research results and integrative metabolomics and transcriptomics, 14 selected key genes were analyzed and verified by quantitative real-time PCR (qRT–PCR). The findings are intended to provide theoretical support for analyzing the biosynthetic pathways of flavonoids and for elucidating the molecular mechanisms of camellia oil quality formation.

## 2. Results

### 2.1. MeJA Treatment Increased the Content of Flavonoids in C. vietnamensis

Analysis of the concentration of MeJA treatment of 2, 4, and 10 mM revealed the influence on the yield of total flavonoids in leaves of *C. vietnamensis*. As shown in Figure 1A, the content of total flavonoids increased 1.5-fold compared with CK when treated with 2 mM MeJA for 2 h and 4 mM MeJA for 12 h.

### 2.2. Differentially Accumulated Metabolite Analysis

UHPLC-Q-TOF MS-based plant metabolomics was used to analyze the response of *C. vietnamensis* leaves to MeJA. A total of 1129 metabolites were identified, of which the percentage of lipids and lipid-like molecules was 27.458%, phenylpropanoids and polyketides was 16.032%, organic acids and derivatives was 6.554%, and alkaloids and derivatives accounted for 1.594% (Figure 1B).

Volcano plots were generated for significantly differentially accumulated metabolites (Appendix A). Orthogonal partial least squares discriminant analysis (OPLS-DA) showed that the values of three parameters, R2x, R2y, and Q2, between groups at different time points were all greater than 0.5 (Appendix A). After 7-fold cross-validation, the obtained model parameters showed that the model was dependable and stable. Additionally, the permutation test was used to test the model. The results showed that both the R2 and Q2 of the stochastic model gradually decreased, indicating that the original model was not overfitted and stable (Appendix A). The variable importance for the projection (VIP) obtained by the OPLS-DA model can be used to measure the impact strength and explanatory power of the expression patterns of each metabolite for the classification and discrimination of each group of leaf samples. Eighty-six (2 h) and 94 (48 h) significantly differentially accumulated metabolites were screened (Figure 1C and Table 1). *C. vietnamensis* had 35 significantly differentially accumulated metabolites at different times (2, 48 h) in response to MeJA treatment (Figure 1C and Appendix A). Toxin HT 2, [1-(3,4-dihydroxy-5-methoxyphenyl)-7-(3,4-dihydroxyphenyl)heptan-3-yl] acetate, senkyunolide A, dihydropalutropone, 5-methyl-1,2,3,4-tetrahydronaphthalene, cinnamaldehyde and other metabolites were significantly upregulated 2 h after MeJA treatment. Metabolites, such as flemiphilippinin A, forsythoside G, (2R,3E)-4-[(1S,6R)-1-hydroxy-2,2,6-trimethyl-4-oxocyclohexyl]-3-buten-2-yl 6-O-beta-D-xylopyranosyl-beta-D-glucopyranoside, leupeptin, and diadinoxanthin A, were significantly upregulated after 48 h of treatment, whereas jasmonic acid, gamma-tocotrienol, methyl jasmonate, and perillene metabolites were upregulated at both time points after treatment (Appendix A).

### 2.3. Hierarchical Cluster Analysis

The results showed that metabolites had different expression patterns at different time points after MeJA treatment, and could be roughly divided into 5 clusters (Figure 1D). The main metabolites in cluster 1 were abietic acid, beta-carotene, kaempferol-7-neohesperidoside, L-glutathione (oxidized form), quercitrin and kaempferol, which exhibited a downward trend after MeJA treatment. The expression of metabolites, such as 1,6-digalloyl-beta-D-glucopyranose, 4-(3,4-dihydroxyphenyl)-5-[(6-O-beta-D-xylopyranosyl-beta-D-glucopyranosyl)oxy]-7-methoxycoumarin, cyanidin 3-(2G-glucosylrutinoside) and camellikaempferoside A, in cluster 2 decreased after 48 h treatment. The expression of metabolites classified in cluster 3 gradually increased after MeJA treatment, and metabolites included fatty acids, such as trans-vaccenic acid, linoleic acid, and palmitic acid. The expression of cluster 4 metabolites increased after MeJA treatment for 48 h, and the metabolites included dioscin, (2R,3E)-4-[(1S,6R)-1-hydroxy-2,2,6-trimethyl-4-oxocyclohexyl]-3 -buten-2-yl 6-O-beta-D-xylopyranosyl-beta-D-glucopyranoside, teviolbioside, flemiphilippinin A and other saponins, flavonoids and alkaloids. The metabolites in cluster 5, including aspartic acid, vitamin C, vitamin E and cinnamaldehyde, exhibited increased expression after 2 h of treatment but decreased expression after 48 h (Appendix A).

In addition, 104 flavonoids (including 12 isoflavonoids, 6 flavonoids, 2 o-methylated flavonoids, 73 flavonoid glycosides, 5 biflavonoids and polyflavonoids, 4 flavans, 1 pyranoisoflavonoid and 1 hydroxyflavonoid) and 74 fatty acyls (including 39 fatty acids and conjugates, 23 fatty acyl glycosides, 1 fatty acyl thioester, 5 fatty alcohols and 6 fatty amides) were identified in the metabolite profiles. These metabolites showed different expression patterns (increased or decreased expression) after MeJA treatment (Appendix A).

### 2.4. Kyoto Encyclopedia of Genes and Genomes (KEGG) Analysis

Among the differentially accumulated metabolites of *C. vietnamensis* leaves in response to MeJA, a total of 162 were aligned to the KEGG database (Appendix A). Among them, three differentially accumulated metabolites were annotated in the phenylpropanoid biosynthesis (ko00940), flavonoid biosynthesis (ko00941), and flavone and flavonol biosynthesis (ko00944) pathways, namely, cinnamaldehyde, kaempferol and quercitrin. Interestingly, three metabolites annotated to the flavonoid biosynthesis pathway were all differentially expressed metabolites noted after 2 h of MeJA treatment (Appendix A).

Comparative analysis using the KEGG database revealed that there were generally more metabolites enriched in metabolic pathways in the 2 h comparison group, and most of them tended to be upregulated. In the 48 h comparison group, metabolites enriched in unsaturated fatty acid biosynthesis were more abundant (Figure 2A). The overall change analysis revealed the metabolites in metabolite biosynthesis pathways (e.g., clavulanic acid biosynthesis, biosynthesis of various secondary metabolites and monobactam biosynthesis pathway) were upregulated, whereas those in the flavone and flavonol biosynthesis pathways were downregulated overall in the 2 h group (Figure 2B). Fewer metabolites were present in the pathways for the 48 h group compared with the 2 h group, and the metabolites involved in unsaturated fatty acid biosynthesis and linoleic acid metabolism upregulation were prominent (Figure 2B).

Interestingly, metabolites related to unsaturated fatty acids (ko01040), such as gamma-linolenic acid and linoleic acid, were upregulated at 2 h and/or 48 h and belonged to the classes including fatty acyls, fatty acids and conjugates, linoleic acids and derivatives (Appendix A).

### 2.5. Transcriptome Sequencing and Gene Functional Annotation

A total of 795,477,852 clean read fragments were obtained by sequencing the transcriptome of the leaf samples of *C. vietnamensis* treated with MeJA. Trinity was used to assemble the leaf samples to obtain the reference sequence of the differential transcriptome. A total of 148,844 genes were sequenced (Table 2). The N50 length was 1276 bp, and the maximum and minimum gene lengths were 15,639 bp and 201 bp, respectively.

After the unigenes of the *C. vietnamensis* transcriptome were assembled, they were searched against four databases (Nr, KOG, KEGG and SwissProt). In total, 74,465 unigenes were annotated (Figure 3A). The unigene sequences assembled with the NR database were used to identify the species corresponding to homologous sequences and calculate the number of homologous sequences in each species. A total of 70,919 unigenes were annotated, of which 43,344 identical unigenes corresponded to *C. sinensis*, accounting for 61.1%. There were 6330 identical unigenes of *Actinidia chinensis*, accounting for 8.9%. In addition, *C. vietnamensis* had similar sequences with *Vitis vinifera*, *Rhodamnia argentea* and wild carrot (*Daucus carota*), accounting for 3%, 2% and 1.6%, respectively (Figure 3B). By annotating all the unigenes in the KEGG database for comparison, the total number of annotated unigenes was 19,031. All unigenes were finally classified into five classes according to their pathway information (Figure 3C). By annotating all the unigenes in the KOG database for comparison, the total number of annotated unigenes was 32,632. All unigenes were finally classified into 25 classes according to their functional information (Appendix A). Based on GO functional analysis, 53 branches could be further refined. There were 24 branches in the biological process category, involving 127,470 unigenes; among them, 21,796 unigenes were annotated to metabolic processes, accounting for 17.10%. Twelve branches were in the molecular function category, which included 37,225 unigenes, and 17 branches were in the cell component category, which included 81,257 unigenes (Appendix A).

### 2.6. Identification and Analysis of Differentially Expressed Genes

Taking the *C. vietnamensis* leaves treated with sterile water under the same conditions as the control, the differentially expressed genes were analyzed after 10 mM MeJA treatment for 2, 24, and 48 h. Based on the FPKM value of each gene, 26,335 genes with FDR < 0.05 and |log2FC| > 1 were screened as significantly differentially expressed (Figure 4A). There were 741 differentially coexpressed genes at 2, 24, and 48 h after MeJA treatment. Among them, 15,702 unigenes (6301 upregulated and 9401 downregulated genes) were identified in the 2 h comparison group. A total of 1958 unigenes (938 upregulated and 1020 downregulated genes) were identified in the 24 h comparison group, and 8675 unigenes (3735 upregulated and 4940 downregulated genes) were identified in the 48 h comparison group (Figure 4B). This finding indicated that there were more downregulated than upregulated unigenes in *C. vietnamensis* under MeJA treatment.

The results showed that the differentially expressed genes related to biological processes in the 2 h comparison group were mainly enriched in ‘metabolic process (3713)’, ‘single-organism process (3173)’ and ‘cellular process (3666)’. Those genes related to molecular functions were mainly enriched in ‘catalytic activity (3112)’, and those genes related to cellular components were mainly enriched in ‘cell part (3003)’. In the 24 h comparison group, the differentially expressed genes related to biological processes were mainly enriched in ‘metabolic process (792)’, those related to molecular functions were enriched in ‘catalytic activity (608)’, and those related to cell components were mainly enriched in ‘cell part (663)’, ‘intracellular (632)’ and “organelle (566)”. The differentially expressed genes in the 48 h comparison group related to biological processes were mainly enriched in ‘metabolic process (2960)’ and ‘cellular process (2924)’, which were related to ‘catalytic activity (2264)’ and ‘cell part (2527)’ (Appendix A). MeJA treatment obviously affected the metabolic process and catalytic activity of *C. vietnamensis*. The effect was more significant after 2 h and 48 h of treatment.

The function of these differentially expressed genes could be further understood based on KEGG analysis. The 2 h, 24 h and 48 h comparison groups revealed 1624, 331 and 1357 genes mapped to the KEGG database, involving 134, 99, and 130 pathways, respectively. The KEGG Orthology (KO) enrichment circle diagram of differentially expressed genes showed that ‘metabolic pathways (ko01100)’ and ‘biosynthesis of secondary metabolites (ko01110)’ were mainly enriched in the three comparison groups. In addition, plant hormone signal transduction (ko04075) and carbon metabolism (ko01200) were enriched by differentially expressed genes (DEGs) at 2 and 48 h, respectively (Figure 4C).

### 2.7. MapMan Analysis of Differentially Expressed Genes

Further insights into the data can be achieved by analyzing differentially expressed transcripts associated with the stress response at the various time points of MeJA treatment (Figure 5). At 2 h of MeJA treatment, more transcripts related to proteolysis, signaling, cell wall and secondary metabolism responded to MeJA among all the differentially expressed transcripts. However, differentially accumulated transcripts related to this function decreased at 24 h and increased again after 48 h. A total of 352 transcripts were enriched in the secondary metabolic pathway, of which 251 transcripts were upregulated and 101 were downregulated. Sixteen of the 251 upregulated transcripts were more strongly upregulated [log2(FC) > 10], including terpenoids, phenylpropanoids, sulfur-containing glucosinolates, wax, anthocyanins, dihydroflavonols, flavonols, isoflavones, and simple phenols. Three of the 101 downregulated transcripts, involving alkaloid-like, sulfur-containing, and dihydroflavonols, were more strongly downregulated [log2(FC) < −10]. The strongly upregulated genes involved in the metabolism of flavonoids after treatment for 2 h were unigene0095353 (anthocyanins), unigene0133783 (anthocyanins), unigene0095353 (dihydroflavonols), unigene0126211 (dihydroflavonols), unigene0126211 (flavonols), unigene0126211 (flavonol 3-O-glycosyltransferase), and unigene0083701 (isoflavone reductase). The most strongly downregulated gene was unigene0048374 (dihydroflavonol 4-reductase). After 24 h of treatment, the differentially expressed transcripts enriched in secondary metabolic pathways were reduced to 100, of which 58 were upregulated and 42 were downregulated, and 2 transcripts involving flavonoids were more strongly upregulated [unigene0140264 (anthocyanins), unigene0070848 (flavonols)]. In contrast, 207 transcripts were enriched in secondary metabolic pathways after 48 h of treatment, of which 100 transcripts were upregulated and 107 were downregulated. Moreover, 9 transcripts involved in tocopherol biosynthesis, sulfur-containing glucosinolate synthesis, dihydroflavonols, flavonols and simple phenols showed strong upregulation, and 3 transcripts involved in lignin biosynthesis, anthocyanins and dihydroflavonols showed a strong downregulated trend. Among the strongly upregulated genes, unigene0070848 and unigene0054738 were both involved in the flavonoid biosynthesis pathway, whereas the strongly downregulated genes unigene0064801, unigene0041807 and unigene0107224 were involved in the synthesis of anthocyanins and dihydroflavonol. Unigene0070848 was strongly upregulated at 24 and 48 h and may be a candidate gene involved in flavonoid biosynthesis. And the results found that unigene0070848 exhibited a regulatory relationship with unigene0070063, unigene0069332 and unigene0130907 (Appendix A).

### 2.8. Coexpression Network Analysis of Metabolomic and Transcriptomic Data

To further identify the coexpression gene module of *C. vietnamensis* leaves in response to MeJA treatment, WGCNA was performed using the expression abundance of 14 metabolites in subject leaves as phenotypic traits and transcriptome data, including JA, MeJA, and 12 metabolites involved in flavonoid biosynthesis. The screened 14,792 genes were used to test the power value. The results showed that when the power was greater than 10 and less than 20, the correspondence between the correlation coefficient and the average connectivity could be guaranteed (Appendix A) with a total of 17 differently colored modules presented in the form of a cluster dendrogram (Figure 6A). Based on the relative contents of the metabolites we focused on, a significant positive correlation was observed between the blue module containing 2766 genes and the expression patterns of JA, MeJA, and cinnamaldehyde (absolute relative coefficient > 0.90, *p* value ≤ 0.01) (Figure 6B).

Moreover, KEGG analysis in this module was carried out. Among them, 283 unigenes were annotated into metabolic pathways (ko01100), accounting for 59.1%; 199 unigenes were annotated to the biosynthesis of secondary metabolites (ko01110), accounting for 41.45%; and the phenylpropanoid biosynthesis pathway (ko00940) was associated with 45 annotated unigenes (Appendix A). Among them, for the genes in the blue module, expression peaked 2 h after MeJA treatment (Appendix A). Notably, there were 2, 4, and 5 unigenes annotated to *PAL*, *4CL*, and *CCR*, respectively. The hub genes were also further mined in each module (Figure 6C). One subtilisin-like transcript, unigene0127420, and glutathione S-transferase L3-like, unigene0002782, had the strongest connectivity, which was arranged by weight value among the top 100 connectivity pairs. The relationship pair also included two transcripts (unigene0132244 and unigene0008507) annotated as TFs, which belong to the MYB and WRKY TF families, respectively. In addition, two MYC2 TFs were found in this module, in which one MYC2 (unigene0138265) was upregulated 69-fold after MeJA treatment compared to the control level, and another MYC2 (unigene0092017) was upregulated 3.7-fold. Nine bHLHs were also found, with multiple bHLHs increasing more than 7-fold after 2 h of treatment. In addition, 9 ERF, 2 LBD, and 2 bHLH TF family members were also annotated in the blue module.

### 2.9. Combined Analysis of Differentially Accumulated Metabolites and Differentially Expressed Genes Related to Flavonoids

To better understand the regulatory network of *C. vietnamensis* leaves in the process of flavonoid metabolism, the correlation analysis between differentially expressed genes and metabolites annotated in the flavonoid metabolism pathway was performed. Based on MapMan analysis results, the differentially expressed genes related to flavonoids were focused on, among which there were 35 coexpressed DEGs (Figure 6D and Appendix A). Thirty-five coexpressed differentially expressed genes and differentially accumulated metabolites (kaempferol and quercitrin, annotated to flavonoid biosynthesis and flavone and flavonol biosynthesis) were plotted on a correlational interaction network (Appendix A). Finally, a candidate gene (unigene0026139) designated F3H was identified as possibly being involved in the biosynthesis of flavonoids.

Based on the reported plant flavonoid metabolic pathways, we further integrated our metabolomic and transcriptomic datasets and mapped the proposed flavonoid biosynthesis pathway of *C. vietnamensis* (Figure 7). In this pathway map, it was observed that MeJA activated the expression of some genes in the flavonoid pathway (*PAL*, *4CL*, *CCR*, *4CH*, *F3H*, *FLS*, etc.) and resulted in the accumulation of upstream metabolites (such as cinnamaldehyde). In addition, MeJA stimulation also downregulated the expression of some genes downstream of the flavonoid biosynthesis pathway (such as *CYP75A*) and inhibited the accumulation of metabolites, such as kaempferol and quercitrin. The expression of a gene is often controlled by multiple TFs, so it is necessary to identify TFs and analyze their functions. At present, an increasing number of studies have shown that the expression of multiple TF families (such as MYB, bHLH, WRKY, AP2/ERF, etc.) is one of the critical mechanisms by which MeJA regulates the synthesis of secondary metabolites. Therefore, TF sequences related to flavonoid and alkaloid biosynthesis were downloaded and performed phylogenetic tree analysis with TFs in our dataset (Appendix A), which were all associated with TFs related to the synthesis of secondary metabolites, such as flavonoids in other species.

### 2.10. Expression Analysis of Key Genes through RT–qPCR Validation

Some key structure genes, such as 14 key enzyme-encoding genes (including *PAL*, *4Cl*, *CHS*, *CHI*, *C4H*, *CCR*, *FLS*, and *UGT* genes) in the proposed flavonoid biosynthesis pathway, were selected for RT–qPCR validation based on the above results for validation of the reliability of the transcriptome data. The expression patterns were generally consistent with the RNA-seq results. Correlation analysis based on the RT–qPCR and transcriptome data also showed a significant correlation with a Spearman correlation coefficient of 0.824, which reflects the accuracy of transcriptome data (Figure 8).

## 3. Discussion

According to reports, proteolysis can play a role in MeJA-induced programmed cell death through cysteine protease and its inhibitors [25]. In addition, plants are activated in response to abiotic or biotic stresses, such as pathogens. For example, reactive oxygen species (ROS), some pathogenesis-related (PR) proteins and other secondary metabolites are produced and accumulate [26], whereas MeJA-induced synthesis of PR-proteins and other stress proteins or secondary metabolites improves the resistance of plants, as previously studied in tomato, sweet pepper [27], peanut [28], other plants, and even the fungus *Ganoderma lucidum* [29]. In this study, the MapMan results also showed that the differentially expressed unigenes were associated with signaling, abiotic stress, proteolysis, PR-proteins, and secondary metabolites (Figure 5). Glutathione-S-transferases (GSTs) widely exist in animals, plants and microorganisms. GSTs play a role in biotic and abiotic stress responses, participating in secondary metabolite synthesis, herbicide detoxification, and signal transduction [30,31]. The results of this study reaffirmed the fact that GSTs respond to stress in plants. GST transcripts were upregulated at each time point after treatment in *C. vietnamensis* (Figure 5) and were clearly responsive to MeJA signaling. A large number of differentially expressed genes related to MVA pathways, glucosinolate, wax, flavonoids, phenylpropanoids, simple phenols, lignin and lignans also confirmed that every metabolic pathway in *C. vietnamensis* was significantly more abundant under the stimulation of MeJA signaling.

Analysis of the transcriptome data revealed that the expression of some key genes on the flavonoids pathway showed a decreasing trend upon MeJA treatment (Figure 7), such as *PAL* (Unigene0008946, Unigene0009565), *4CL* (Unigene0022021, Unigene0066283), *CHI* (Unigene0031065, Unigene0031066), *F3H* (Unigene0143055), and *F3**′5**′H* (Unigene0008455, Unigene0023819). The metabolome results showed that the main metabolites in cluster 1 were beta-carotene, kaempferol-7-neohesperidoside, quercitrin and kaempferol, which exhibited a downward trend after MeJA treatment (Figure 1D). Among them, quercitrin and kaempferol were the products between the flavonoids synthesis pathways. The overall change analysis revealed the flavone and flavonol biosynthesis pathways were downregulated overall in the 2 h group (Figure 2B). To sum up, MeJA treatment affected the expression of some genes of the pathway upstream of flavonoids, leading to a decrease in the content of some intermediate products (precursors for flavonoids synthesis, such as quercitrin and kaempferol); the decrease in the contents of these intermediate products might have affected the expression of downstream genes of the flavonoids synthesis pathway in a short period of time, leading to the decrease in the contents of end products (such as phlorizin, quercitrin) and thus the total flavonoid content.

Genes, including structural and regulatory genes, involved in flavonoid biosynthesis and regulation have been reported in many plants. 14 structural genes and 5 TF genes in *Actinidia arguta* were selected, which involved in the flavonoid biosynthesis pathway [32]. TFs such as bHLH3, bHLH18, MYC2, and MYB44, were involved in the MeJA-mediated biosynthesis of flavonoids in *C. oleifera* seeds [21]. However, there are few studies on the genes involved in flavonoid biosynthesis and regulation in *C. vietnamensis*. In this study, metabolome and transcriptome were carried out in a combined analysis for the discovery of genes involved in flavonoid biosynthesis, thus searching for useful information to analyzing the molecular regulatory network of flavonoids under MeJA treatment in *C. vietnamensis*.

In response to MeJA, it was found that only two differential metabolites, kaempferol (ko00941) and quercitrin (ko00944), were annotated in the flavonoid pathway in the KEGG annotation. The association analysis between these two metabolites and the differentially expressed genes annotated in flavonoid biosynthesis showed that one F3H gene exhibited the strongest correlation with metabolites. Therefore, we hypothesized that it might be involved in the synthesis of flavonoids in *C. vietnamensis*. The expression pattern was significantly upregulated at two time points (i.e., 2 and 48 h) after MeJA treatment compared with CK, but at a very low level. The expression level was negatively correlated with the accumulation of metabolites downstream of the flavonoid metabolic pathway. The expression levels of upstream key genes and metabolites related to flavonoid biosynthesis in this study were promoted, whereas those of downstream genes and metabolites were inhibited. This finding is similar to the results in *Carthamus tinctorius* L. [22]. The WGCNA analysis showed that the gene expression pattern of blue module was similar to that of midnightblue module. The expression of genes in these two modules increased after MeJA treatment for 2 h. Unlike the blue module, the genes in the darkgreen module and the lightcyan module had the highest expression in ck-48 h. And the results observed a significant positive correlation between the genes in the blue module and the expression patterns of JA, MeJA, and cinnamaldehyde. On the contrary, the genes in the darkgreen module and the lightcyan module had a negative correlation with the expression patterns of JA, MeJA and cinnamaldehyde. Such as Unigene0039239 (*PAL*), Unigene0080700 (*4CL*), and Unigene0025733 (*CCR*) showed an almost sixty-fold, fourteen-fold, and twenty-fold expression difference compared to the CK levels at 2 h after treatment, respectively. These were all candidate genes that might be involved in the synthesis of cinnamaldehyde. However, the responses of different plant species to flavonoids were different. For example, Premathilake et al. induced pear calli with MeJA and found that MeJA treatment could upregulate the expression of structural genes in the pear flavonoid biosynthesis pathway with more prominent upregulation of anthocyanin- and proanthocyanidin-related genes [33]. JA had a stimulatory effect on genes related to flavonoid and anthocyanin biosynthesis in *A. thaliana*, including *AtPAL* and *AtCHS*, but their relative expression levels were not very high [34]. *UGTs* are widely involved in the last step in the synthesis of secondary metabolites. Current studies have shown that although most *UGTs* have specificity for the selection of catalytic substrates, some of substrates are extensive and complex and are typically flavonoids and terpenoids [35]. Achnine et al. noted that *UGT71G1* could catalyze flavonoid through functional verification [36]. Therefore, *UGT* genes screened in our study might also have a variety of catalytic substrates (such as triterpenoid saponins and flavonoids) which need further functional analysis.

In addition to structural genes, TFs are important in flavonoid biosynthesis [37]. Three modules responsible for flavonoid accumulation were identified by WGCNA, in which TFs such as bHLH3, bHLH18, MYB44, MYB86, WRKY26, and WRKY32 were predicted to participate in flavonoid biosynthesis [21]. Eight MYBs (EpMYB108, EpMYB111, EpMYB84, EpMYB85, EpMYB78, EpMYB115, EpMYB86, and EpMYB76) were selected which related to anthocyanin biosynthesis [6]. In the dataset, TFs were screened that clustered onto a branch of flavonoid-related TFs from other species. The three MYBs, unigene0077915, unigene0049565, and unigene0051487, had the same expression pattern as cinnamaldehyde, which suggested that these three MYB TFs played a positive regulatory role in cinnamaldehyde biosynthesis. Two WRKYs (unigene0005872 and unigene0123624) might be involved in the accumulation of kaempferol and quercitrin (Appendix A). The results suggest that these MYBs may have the potential ability to regulate flavonoids biosynthesis, but further validation is required

## 4. Materials and Methods

### 4.1. Materials, MeJA Treatment and Leaf Sample Preparation

Healthy one-year-old grafted *C. vietnamensis* ‘Wanhai No. 4′ seedlings (when the shoots were more than 10 cm) were grown in a camellia nursery greenhouse of Hainan University, China (19°30′28″ N, 109°29′45″ E). MeJA-treated leaves samples and blank controls (CKs) were prepared according to the methods reported by Shi [38] and Wang [27]. Formulation of MeJA solutions at different concentrations: 2 mM MeJA (0.1 mL MeJA dissolved in 4 mL ethanol absolute and brought to 200 mL with ultrapure water); 4 mM MeJA (0.2 mL MeJA dissolved in 4 mL ethanol absolute and brought to 200 mL with ultrapure water); 10 mM MeJA (0.5 mL MeJA dissolved in 4 mL ethanol absolute and brought to 200 mL with ultrapure water); CK (4 mL ethanol absolute brought to 200 mL with ultrapure water). Leaf tables of *C. vietnamensis* annual seedlings (25 *C. vietnamensis* annual seedlings in each treatment group) were separately sprayed at 8 am with a spray pot filled with different concentrations of MeJA solution until dripping, respectively, with a plastic film barrier between each treatment group. Then leaf samples were taken at 0, 2, 6, 12, 18, 24, 48 and 72 h [27,39] for detection of the content of total flavonoids.

Then, a total of six fresh leaf samples from each seedling were treated with water (blank control [CK]) or a 10 mM MeJA solution at the time points of 2, 24, and 48 h, quickly placed into liquid nitrogen and then stored at −80°C for metabolomic (5 biological samples for each timepoint and group) and transcriptomic (3 biological samples for each timepoint and group) analyses.

### 4.2. Determination Conditions, Data Processing and Multivariate Statistical Analysis of UHPLC-QTOF/MS Results

Analyses were performed using an ultra-high performance liquid chromatograph (1290 Infinity LC, Agilent Technologies, Santa Clara, CA, USA) coupled to a quadrupole time-of-flight mass spectrometer (AB SCIEX TripleTOF 6600, Framingham, MA, USA) at Shanghai Applied Protein Technology Co., Ltd. (Shanghai, China). The leaf samples were separated by an Agilent 1290 Infinity LC ultra-performance liquid chromatograph on a C-18 column; the column temperature was 40 ℃. The flow rate was set at 0.4 mL/min, and the injection volume was 2 μL. Mobile phase A was 25 mM ammonium acetate + 0.5% formic acid in water, and mobile phase B was methanol. The gradient elution procedure was as follows: 0–0.5 min, 5% B; then, B changed to 100% linearly from 0.5 to 10 min; from 10–12 min, B was maintained at 100%; from 12.0 to 12.1 min, B changed linearly from 100% to 5%; from 12.1–16 min, B was maintained at 5%. During the whole analysis, the leaf sample was placed in an automatic sampler at 4 ℃. To avoid the influence caused by the fluctuation of the instrument, a random sequence was used for the analysis of leaf samples. Quality control (QC) samples are inserted into the leaf sample queue to monitor and evaluate the stability and reliability of the data. Electrospray-ionization (ESI) source conditions were defined as follows: 60 for ion source gas 1, 60 for gas 2 69, 30 for CUR, 600 °C source temperature, and ion spray floating point (ISVF) ± 5500 V. In MS-only acquisition, the *m*/*z* range and TOF MS scan were from 60 to 1000 Da and at 0.20 s/spectra. The auto MS/MS data were acquired with an *m*/*z* range from 25 to 1000 Da, and the accumulation time for the leaf sample ion scan was 0.05 s/spectra. The leaf sample ion scan with information-dependent acquisition (IDA) was performed on the high-sensitivity mode [40].

The following parameters were established: centWave *m*/*z* of 25 ppm, peak width of c (10, 60), prefilter of c (10, 100), bw of 5, mzwid of 0.025, and minfrac of 0.5. Metabolite identification was performed by an in-house (Shanghai Applied Protein Technology) database established with available authentic standards for compound identification [41,42]. The processed data were normalized to total peak intensities and analyzed using the R package for further multivariate analysis. Differentially abundant metabolites with a variable importance in projection (VIP) value exceeding 1 were further subjected to Student’s t test (*p* values < 0.05) [40]. Hierarchical clustering was performed via the OmicShare website (https://www.omicshare.com; accessed on 1 February 2022).

### 4.3. Transcriptome Sequencing, Gene Expression Analysis, Annotation and Differential Gene Analysis

Total RNA was isolated from the leaf samples with a TRIzol reagent kit (Invitrogen, Carlsbad, CA, USA). Then, eukaryotic mRNA was enriched using oligo(dT) beads. Prokaryotic mRNA was enriched by removing rRNA with a Ribo-ZeroTM Magnetic Kit (Epicenter, Madison, WI, USA) and fragmented into short fragments by fragmentation buffer followed by double-strand cDNA synthesis. Finally, the ligation products were size selected, polymerase chain reaction (PCR) amplified, and sequenced using an Illumina

NovaSeq 6000 by Gene Denovo Biotechnology Co. (Guangzhou, China). Then, high-quality clean reads were filtered by fastp [43] (version 0.18.0). The parameters were as follows: (1) reads containing adapters were removed; (2) reads containing more than 10% unknown nucleotides (N) were removed; and (3) low-quality reads containing more than 50% low-quality (Q-value ≤ 20) bases were removed. Transcriptome de novo assembly was carried out with the short read assembly program Trinity [44]. Fragments per kb per million reads (FPKM) values were used to determine the unigene expression level [45]. The data were then annotated to four databases (Non-Redundant (NR), Eukaryotic Orthologous Groups (KOG), Kyoto Encyclopedia of Genes and Genomes (KEGG), and SwissProt), and protein functional annotations were obtained according to the best alignment results.

Differential RNA expression analysis of the two leaf sample groups was performed using DESeq2 package (version 1.36.0) (http://www.bioconductor.org/packages/release/bioc/html/DESeq2.html; accessed on 1 May 2021) [46] (two leaf samples were analyzed with edgeR [47]). A false discovery rate (FDR) under 0.05 and absolute fold change ≥ 2 were used as the thresholds for the selection of differentially expressed genes.

### 4.4. MapMan, WGCNA and NCBI Blast + Analysis

Differentially expressed transcripts were mapped onto overviews of biotic stress and secondary metabolism using MapMan software (version 3.6.0RC1, Virginia Tech, Blacksburg, VA, USA) (https://mapman.gabipd.org/; accessed on 1 February 2022). The sequence files of all transcripts in the transcriptome were annotated by a website (http://www.plabipd.de/portal/mercator-sequence-annotation; accessed on 1 May 2021), and the resulting file was the mapping file used in this MapMan analysis. WGCNA [48] can be used to evaluate the gene expression patterns of multiple samples. WGCNA was performed with the R package and Cytoscape software (v3.6.0), with a power value of 10, module similarity of 0.9 and minimum number of genes of 50.

### 4.5. Verification of qPCR Analysis

RT–qPCR was carried out on a Roche LightCycler 96 (F. Hoffmann-La Roche Ltd., Basel, Switzerland) using a two-step method plus melting curve detection with 40 cycles. Primers were designed by Primer 5.0 (Premier Biosoft, Palo Alto, CA, USA) (Appendix A). The relative expression levels of genes were normalized to those of actin and glyceraldehyde-3-phosphate dehydrogenase (GAPDH) and analyzed using the 2-ΔΔCt method.

## 5. Conclusions

In summary, this study focused on the key molecular mechanisms underlying metabolic regulation (related to flavonoids) in *C. vietnamensis* based on metabolomics and transcriptomics technologies under MeJA treatment. Integrated metabolomic and transcriptomic analyses showed that MeJA positively regulated genes related to flavonoid biosynthesis in *C. vietnamensis* leaves, and multiple enzyme-encoding genes were screened that might be related to different flavonoids. Furthermore, 1 *CHS*, 1 *CHI*, 3 *CCR* and 1 *FLS* were identified as the most promising genes for further developing breeding and research strategies. The studies provide new insights and data that support the molecular regulatory network of flavonoids under MeJA treatment in *C. vietnamensis*

## Figures and Tables

**Figure 1 ijms-23-09370-f001:**
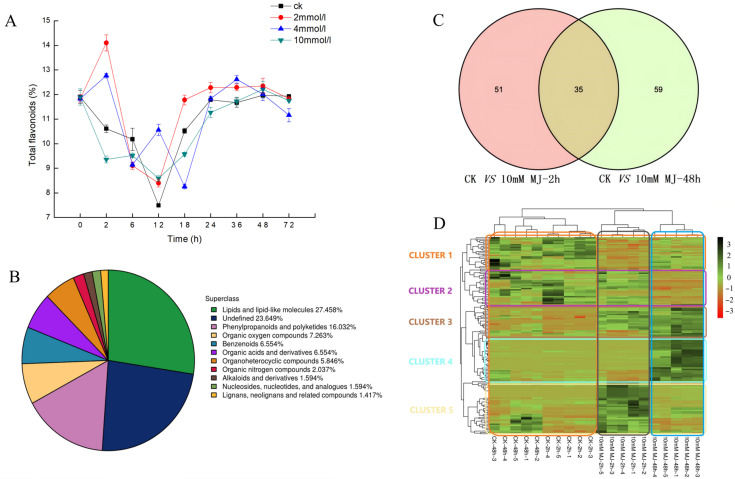
(**A**): Dynamic changes in the content of total flavonoids in *C. vietnamensis* under different concentrations of MeJA treatment. (**B**): Quantitative proportions of identified metabolites in each chemical classification. (**C**): Venn diagram of significantly differentially accumulated metabolites between groups (‘ck vs. 10 mM MJ—2 h’ and ‘ck vs. 10 mM MJ—48 h’). (**D**): Heatmap of significantly differentially accumulated metabolites of *C. vietnamensis* in response to MeJA at different time points (2 and 48 h). Red to green indicates metabolite expression from low to high.

**Figure 2 ijms-23-09370-f002:**
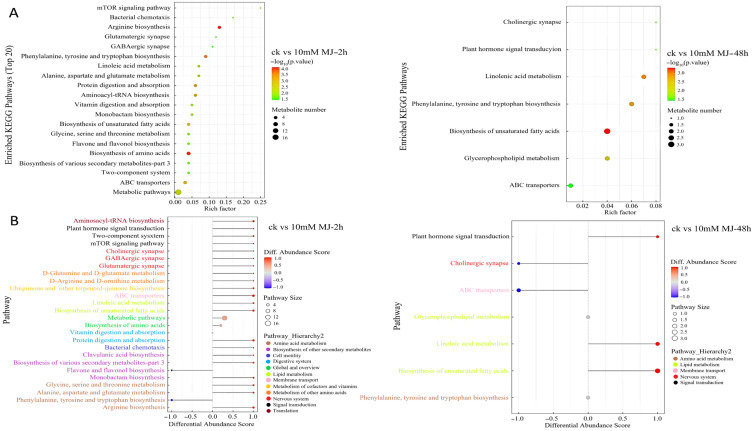
KEGG pathway enrichment of differentially accumulated metabolites in *C. vietnamensis* leaves in response to MeJA for different comparison groups. (**A**): KEGG pathway enrichment of CK vs. 10 mM MJ—2 h and CK vs. 10 mM MJ—48 h. Each bubble represents a metabolic pathway (the top 20 with the most significance were selected according to the *p* value). The size of the bubble represents the number of metabolites in the pathway with topology analysis. The color of the bubble represents the *p* value of the enrichment analysis (i.e., −log10 *p* value); the darker the color, the smaller the *p* value, and the more significant the enrichment degree. (**B**): Differential abundance score map for all differential metabolic pathways. The *Y*-axis in the figure represents the name of the differential pathway, and the *X*-axis coordinates represent the differential abundance score (DA score). The DA score is the overall total change in all metabolites in the metabolic pathway. A score of 1 indicates that all identified metabolites in the pathway exhibit a trend of upregulated expression, whereas −1 indicates a trend of downregulated expression. The darker the red, the more likely the overall expression of the pathway is to be upregulated. The darker the blue, the more likely the overall expression of the pathway is downregulated.

**Figure 3 ijms-23-09370-f003:**
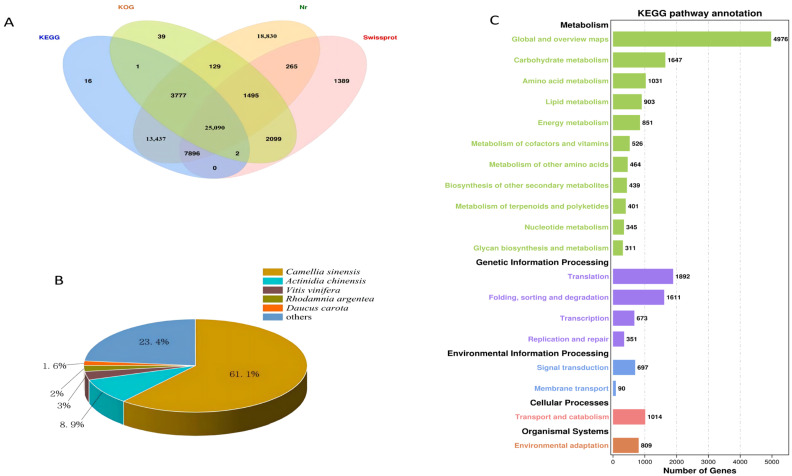
(**A**): Venn diagram annotated using four datasets. (**B**): Species distribution for all unigenes in the transcriptome in NR. (**C**): KEGG pathway classification.

**Figure 4 ijms-23-09370-f004:**
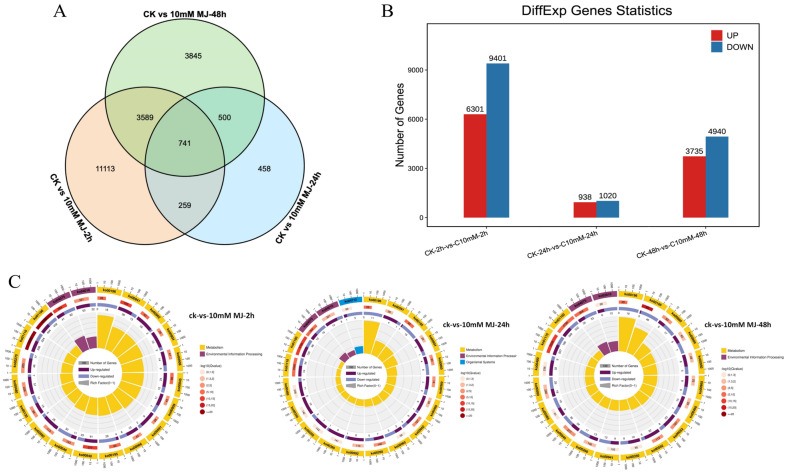
Identification and expression analysis of differentially expressed genes. (**A**): Venn diagram of differentially expressed genes. (**B**): Histogram of differentially expressed genes. (**C**): KO enrichment circular plot of differentially expressed genes. The top 20 pathways were enriched in the first circle with the coordinate ruler of the number of genes noted outside the circle. Different colors represent different classes. The number of pathways and the Q value in the background are reported in the second circle. The more genes, the longer the bar, the smaller the Q value, and the redder the color. A bar graph showing the ratio of upregulated and downregulated genes is shown in the third circle, in which dark and light purple represent the ratio of upregulated genes and downregulated genes, respectively. The RichFactor value of each pathway is shown in the fourth circle (the number of differentially expressed genes in the pathway divided by all the numbers).

**Figure 5 ijms-23-09370-f005:**
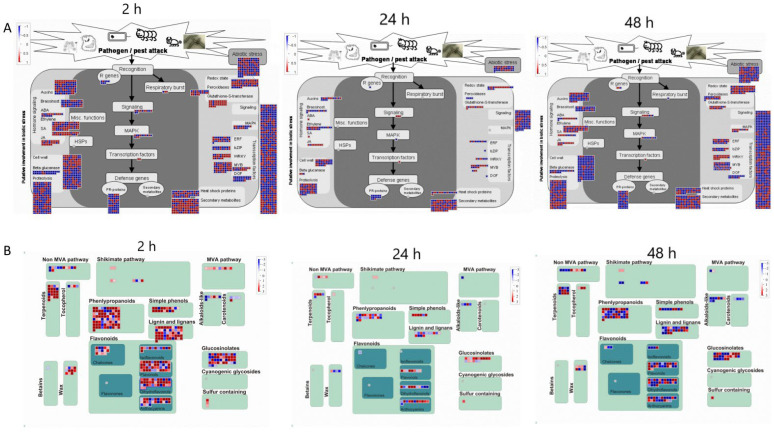
MapMan analysis of differentially expressed genes of *C. vietnamensis* in response to MeJA. (**A**): CK vs. 10 mM MeJA differentially expressed genes associated with stress response. (**B**): CK vs. 10 mM MeJA differentially expressed genes associated with secondary metabolism.

**Figure 6 ijms-23-09370-f006:**
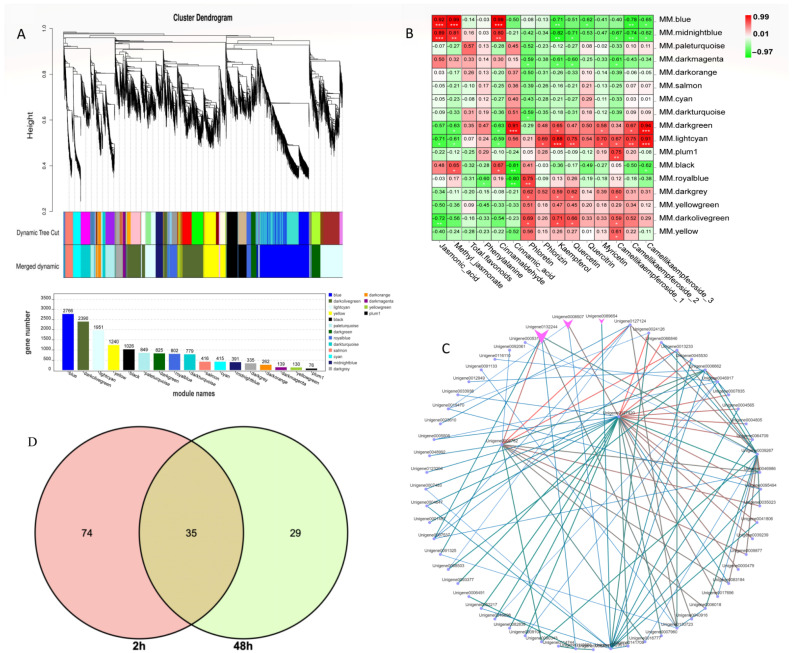
WGCNA of metabolites and differentially expressed genes. (**A**): Differential gene hierarchical module cluster diagram and the number of genes in each module. Genes with similar expression patterns were classified into the same module, and the branches of the clustering tree were cut and differentiated to generate different modules. Each color represents a module, and gray indicates genes that could not be assigned to any one module. The modules were further merged. (**B**): Module-trait relationship of 29 metabolites with 17 gene modules. The Pearson correlation coefficient was used to plot the eigenvalues of the module and the physiological data. Red represents a positive correlation, and green represents a negative correlation. The darker the color, the stronger the correlation. * signifificant at level 0.05, ** signifificant at level 0.01, *** signifificant at level 0.001. (**C**): Cytoscape of the top 100 relationship pairs of connectivity in the module. Points represent genes, and lines represent a regulatory relationship between two points. The darker and larger the node color and line, the stronger the connectivity. (**D**): Venn of differential genes annotated on the flavonoid metabolic pathway.

**Figure 7 ijms-23-09370-f007:**
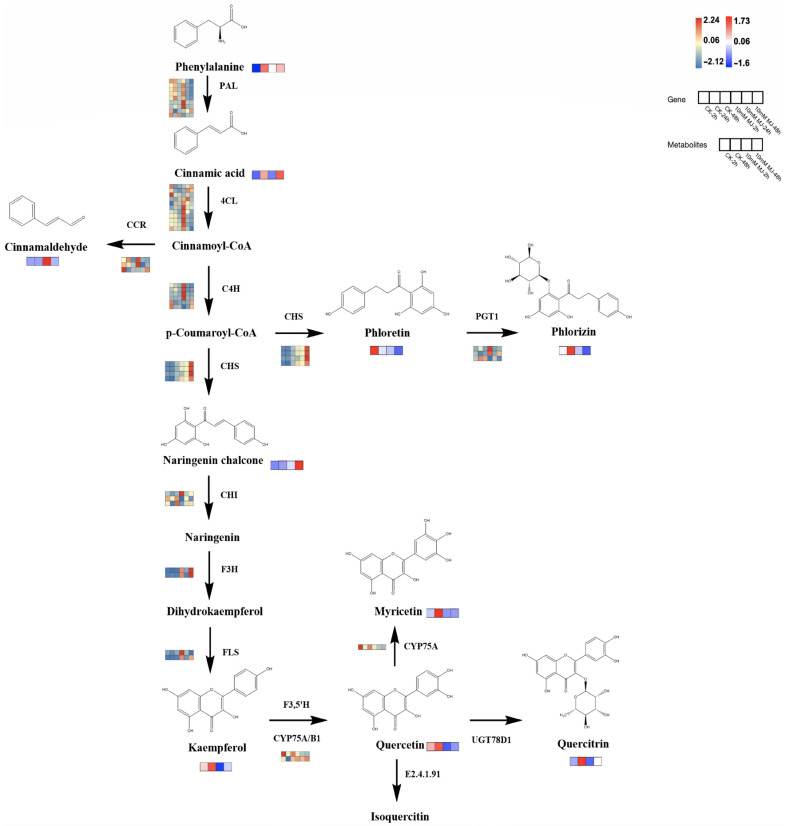
Proposed biosynthetic pathways of flavonoids in *C. vietnamensis.* Blue–yellow–red represents transcripts from downregulated to upregulated, and blue–white–red represents metabolites from downregulated to upregulated.

**Figure 8 ijms-23-09370-f008:**
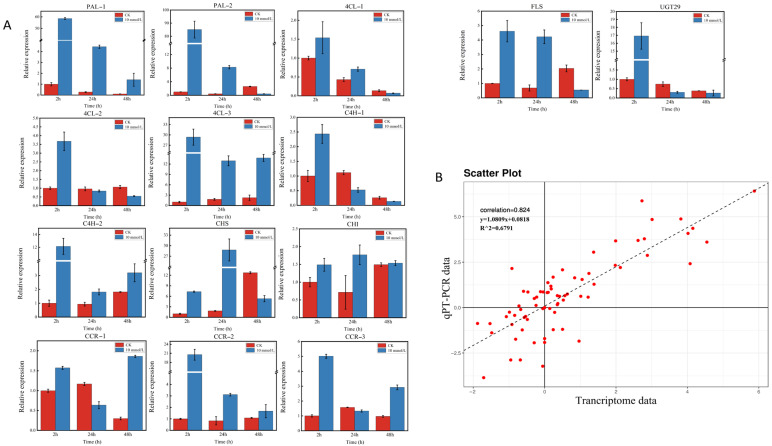
Expression pattern profiles of structural genes related to flavonoid biosynthesis. (**A**): The relative expression levels of the key structural genes in flavonoid biosynthesis pathways. (**B**): Scatter plot and linear regression based on qRT–PCR and transcriptome data. The correlation coefficient was calculated using the Spearman correlation method.

**Table 1 ijms-23-09370-t001:** Differentially accumulated metabolites of *C. vietnamensis* leaves in response to MeJA in the negative and positive ion modes.

	CK vs. 10 mM MeJA
2 h	
More abundant	59
Less abundant	27
48 h	
More abundant	63
Less abundant	31

Note: Significantly differentially accumulated metabolites were screened by using OPLS-DA VIP > 1 and *p* value < 0.05 as the screening criteria. Among them, differentially accumulated metabolites with FC > 1.5 and <0.67 were considered more and less abundant metabolites, respectively.

**Table 2 ijms-23-09370-t002:** Transcriptome analysis data statistics of *C. vietnamensis*.

Unigene Number	GC/%	N50 Number	N50 Length/bp	Length/bp	Min Length/bp	Average Length/bp	Total Assembled Bases
148,844	38.92	24,310	1276	15,639	201	776	115,572,751

## Data Availability

The datasets in this study were uploaded to NCBI, with the accession number PRJNA825399 (https://www.ncbi.nlm.nih.gov/sra/PRJNA825399; accessed on 11 April 2022).

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
