# Peer review of "Integrative Metabolome and Transcriptome Analysis Reveals the Regulatory Network of Flavonoid Biosynthesis in Response to MeJA in Camelliavietnamensis Huang"

_ijms, 2022, doi:10.3390/ijms23169370_

Round 1

Reviewer 1 Report

The MS is devoted to the study of the flavanoids  formation in a Camellia  plant during exogenous treatment with jasmonates.

The authors obtained interesting results that are of interest for modern plant biology.

There are some comments.

There is not a word about processing with MJ in the title, it needs to be corrected.

In the methods several concentrations of MJ were given, but only one is given in the results (expression figures), why?

Treatment with MJ ultimately reduces the content of flavanoids, why this happens according to the authors, it is necessary to supplement the text/

Why MJ and not some other hormone was chosen should be explained in the text.

It is necessary to describe in detail the method of treating plants with hormones, as well as whether any sulfactant Tween or   Silwet was used.

It is necessary to accurately indicate the age of the plants.

The exact organ used in the analysis must be specified.

This article needs to be cited.

https://www.frontiersin.org/articles/10.3389/fpls.2022.880227/full

Author Response

Responses addressing Reviewers’ Comments

Reviewer #1: 

Comment 1: There is not a word about processing with MJ in the title, it needs to be corrected.

Response:

Thank you for the valuable suggestion. In accordance with the Reviewer’s comment, we have changed the title into “Integrative metabolome and transcriptome analysis reveals the regulatory network of flavonoid biosynthesis in response to MeJA in Camellia vietnamensis Huang” in the revised manuscript and are marked in red.

Comment 2: In the methods several concentrations of MJ were given, but only one is given in the results (expression figures), why? 

Response:

Thank you for the valuable suggestion. Because the treatment of C. vietnamensis with different concentrations of MeJA was initiated to screen for the MeJA concentrations that have an effect on the secondary metabolites of C. vietnamensis leaves, transcriptome sequencing was then performed. Transcriptome sequencing used a single concentration of treated C. vietnamensis leaf samples, so only this concentration treatment was analyzed in the results of the expression figures.

Comment 3: Treatment with MJ ultimately reduces the content of flavanoids, why this happens according to the authors, it is necessary to supplement the text.

Response:

Thank you for the valuable suggestion. Analysis of the transcriptome data revealed that the expression of some key genes on the flavonoids pathway showed a decreasing trend upon MeJA treatment (Figure 7), such as PAL (Unigene0008946, Unigene0009565), 4CL (Unigene0022021,Unigene0066283), CHI (Unigene0031065, Unigene0031066), F3H (Unigene0143055), F35H (Unigene0008455, Unigene0023819). The metabolome results showed that metabolites had different expression patterns at different time points after MeJA treatment and could be roughly divided into 5 clusters (Figure 1D). The main metabolites in cluster 1 were abietic acid, beta-carotene, kaempferol-7-neohesperidoside, L-glutathione (oxidized form), quercitrin and kaempferol, which exhibited a downward trend after MeJA treatment. Among them, quercitrin and kaempferol were the products between the flavonoids synthesis pathways. The overall change analysis revealed the metabolites in metabolite biosynthesis pathways (e.g., clavulanic acid biosynthesis, biosynthesis of various secondary metabolites and monobactam biosynthesis pathway) were upregulated, whereas those in the flavone and flavonol biosynthesis pathways were downregulated overall in the 2 h group (Figure 2B).

To sum up, MeJA treatment affected the expression of some genes of the pathway upstream of flavonoids, leading to a decrease in the content of some intermediate products (precursors for flavonoids synthesis, such as quercitrin and kaempferol). And the decrease in the contents of these intermediate products might have affected the expression of downstream genes of the flavonoids synthesis pathway in a short period of time, leading to the decrease in the contents of end products (such as phlorizin, quercitrin) and thus the total flavonoid content.

Considering the reviewer’s suggestion, we have made correction on the discussion section in the revised manuscript and are marked in red.

Comment 4: Why MJ and not some other hormone was chosen should be explained in the text.

Response:

Thank you for the valuable suggestion. We were going to want to treat C. vietnamensis with exogenous hormones and stimulated the plant to produce a response, which led to changes in secondary metabolism and secondary metabolites in the plant. As a well-known exogenous inducing factor, MeJA is widely employed to produce bioactive compounds and can affect secondary metabolite content is now a hot spot of research and has been largely validated on other species, such as Camellia sinensis [19] and Carthamus tinctorius L. [22]. MeJA could affect the up-regulation and down-regulation of genes in the flavonoid metabolic pathway in order to promote the production of flavonoid metabolites [22,23]. Consequently, MeJA treatment can be used to identify the genes involved in the biosynthesis of flavonoids. This method has been widely used in preliminary screening of genes related to the biosynthesis of secondary metabolites so far. Therefore, it is meaningful and feasible to analyze the biosynthetic pathways of flavonoids and saponins under MeJA treatment using both transcriptome and metabolome. In accordance with the Reviewer’s comments, we have have made correction on the introduction in the revised manuscript, marked in red.

References

[19] Song, S.; Tao, Y.; Gao, L.; Liang, H.; Tang, D.; Lin, J.; Wang, Y.; Gmitter, F.G. Jr.; Li, C. An Integrated Metabolome and Transcriptome Analysis Reveal the Regulation Mechanisms of Flavonoid Biosynthesis in a Purple Tea Plant Cultivar. Front. Plant Sci. 2022, 13, 880227. doi: 10.3389/fpls.2022.880227

[22] Chen, J.; Wang, J.; Wang, R.; Xian, B.; Pei, J. Integrated metabolomics and transcriptome analysis on flavonoid biosynthesis in safflower (Carthamus tinctorius L.) under MeJA treatment. BMC Plant Biol. 2020, 20(1), 353. doi: 10.1186/s12870-020-02554-6

[23] Yamamoto, R.; Ma, G.; Zhang, L.; Hirai, M.; Yahata, M.; Yamawaki, K.; Shimada, T.; Fujii, H.; Endo, T.; Kato, M. Effects of Salicylic Acid and Methyl Jasmonate Treatments on Flavonoid and Carotenoid Accumulation in the Juice Sacs of Satsuma Mandarin In Vitro. Appl. Sci. 2020, 10, 8916. doi: 10.3390/app10248916

Comment 5: It is necessary to describe in detail the method of treating plants with hormones, as well as whether any sulfactant Tween or Silwet was used.

Response:

Thank you for this valuable suggestion. MeJA-treated leaves samples and blank controls (CKs) were prepared according to the methods reported by Wang [27] and Shi [38]. Formulation of MeJA solutions at different concentrations: 2 mM MeJA (0.1 ml MeJA dissolved in 4 mL ethanol absolute and brought to 200 mL with ultrapure water); 4 mM MeJA (0.2 ml MeJA dissolved in 4 mL ethanol absolute and brought to 200 mL with ultrapure water); 10 mM MeJA (0.5 ml MeJA dissolved in 4 mL ethanol absolute and brought to 200 mL with ultrapure water); CK (4 mL ethanol absolute brought to 200 mL with ultrapure water). Leaf tables of C. vietnamensis annual seedlings (25 C. vietnamensis annual seedlings in each treatment group) were separately sprayed at 8 am with a spray pot filled with different concentrations of MeJA solution until dripping, respectively, with a plastic film barrier between each treatment group. Then leaf samples were obtained at 0, 2, 6, 12, 18, 24, 48 and 72 h [27,39] for detection of the content of total flavonoids. And we did not used any sulfactant Tween or Silwet. In accordance with the Reviewer’s comments, we have have made correction on “4. Materials and Methods” in the revised manuscript, marked in red.

References

[27] Wang, C.Y.; Fung, R.W.M.; Ding, C.K. Reducing chilling injury and enhancing transcript levels of heat shock proteins, pr-proteins and alternative oxidase by methyl jasmonate and methyl salicylate in tomatoes and peppers. V Int. Postharvest Symposium 2005, 682, 481-486. doi: 10.17660/ActaHortic.2005.682.58

[38] Shi, J.; Wang, L.; Ma, C.Y.; Lv, H.P.; Chen, Z.M.; Lin, Z. Aroma changes of black tea prepared from methyl jasmonate treated tea plants. J. Zhejiang Univ-Sci. B. 2014, 15, 313-321. doi: 10.1631/jzus.B1300238

[39] Chen, C.; Liu, F.; Zhang, K.X.; Niu, X.L.; Zhao, H.; Liu, Q.X.; Georgiev, M.I.; Xu, X.H.; Zhang, X.Q.; Zhou, M.L. MeJA-responsive bHLH transcription factor LjbHLH7 regulates cyanogenic glucoside biosynthesis in Lotus japonicus. J. Exp. Bot. 2022, 73, 2650–2665. doi: 10.1093/jxb/erac026

Comment 6: It is necessary to accurately indicate the age of the plants.

Response:

Thank you for this valuable suggestion and we are very sorry for our negligence regarding the citations. The plants were annual grafted seedlings. We have shown in 4.1. of “4. materials and methods”: “Healthy one-year-old grafted C. vietnamensis ‘Wanhai No. 4’ seedlings (when the shoots were more than 10 cm) were grown in a camellia nursery greenhouse of Hainan University, China (19°30′28″N, 109°29′45″E)”. According to the reviewer’s comment, we have made correction in the revised manuscript and are marked in red.

Comment 7: The exact organ used in the analysis must be specified.

Response:

Thank you for this valuable suggestion and we apologize for the unclear language expression. The exact organ used in the analysis was C. vietnamensis leaf. According to the reviewer’s comment, we have made correction in the revised manuscript and are marked in red.

Comment 8: This article needs to be cited.

https://www.frontiersin.org/articles/10.3389/fpls.2022.880227/full

Response:

Thank you for this valuable suggestion and and we are very sorry for our negligence regarding the citations. According to the reviewer’s comment, we have cited this article (the 19th reference) in the revised manuscript and are marked in red.

Reviewer 2 Report

The main goal of this research is very important. The article ,,Integrative metabolome and transcriptome analysis reveals the regulatory network of flavonoid biosynthesis in Camellia vietnamensis Huang” provides new information about molecular mechanism of flavonoid biosynthesis.

The article is well written, the references is well-chosen, most a large part is the publication from 5 years ago, which proves that such relationships have been intensively researched in recent times. I have one suggestion, the figures showing the results are of poor quality and not very legible.

Author Response

Reviewer #2:

Comment : The main goal of this research is very important. The article “Integrative metabolome and transcriptome analysis reveals the regulatory network of flavonoid biosynthesis in Camellia vietnamensis Huang” provides new information about molecular mechanism of flavonoid biosynthesis.

The article is well written, the references is well-chosen, most a large part is the publication from 5 years ago, which proves that such relationships have been intensively researched in recent times. I have one suggestion, the figures showing the results are of poor quality and not very legible.

Response:

Thank you very much for your recognition of our work and this valuable suggestion. According to the reviewer’s comment, the arrangement and clarity of all the figures have been improved. In addition, the resolution was enhanced so the figures could be read in the revised manuscript.